

# Simulations of Moving Effect of Coastal Vegetation on Tsunami Damping

Ching-Piao Tsai, Ying-Chi Chen, Tri Octaviani Sihombing and Chang Lin

Department of Civil Engineering, National Chung Hsing University, Taichung 402, Taiwan

*Correspondence to*: Ching-Piao Tsai (cptsai@nchu.edu.tw)

**Abstract.** A coupled wave-vegetation simulation is presented for the moving effect of the coastal vegetation on tsunami wave height damping. The problem is idealized by solitary wave propagating on a group of emergent cylinders. The numerical model is based on general Reynolds-averaged Navier-Stokes equations associated with renormalization group turbulent closure model by using volume of fluid technique. The general moving object (GMO) model developed in CFD

code Flow-3D is applied to simulate the coupled motion of vegetation with wave dynamically. The damping of wave height and the turbulent kinetic energy dissipation as waves passed over both moving and stationary cylinders are discussed. As comparing with the stationary cylinders, it obtains markedly less wave height damping and turbulent kinetic energy dissipation by the moving cylinders. The result implies that the wave decay by the coastal vegetation might be overestimated if the mangrove vegetation was represented as stationary state.

## 1 Introduction

A huge tsunami in South East Asia caused catastrophic damage and claimed more than 200,000 people in December 2004. Cochard et al. (2008) pointed out that this event has stimulated a debate about the role played coastal ecosystems such as mangrove forests and coral reefs in protecting low-lying coastal area. For example, Baird (2006) questioned the effectiveness of the coastal forests or reefs on the reduction of the damage caused by the tsunami. However, Danielsen et al. (2005)

reported areas with coastal tree vegetation were markedly less damaged than areas without. Iverson and Prasad (2007) also indicated that developed areas were far more likely to be damaged than forested zones. Several studies (Hiraishi and Harada, 2003; Harada and Kawata, 2004; Teh et al., 2009) have shown that tsunami wave energy, heights and velocities were significantly reduced as the wave propagates through mangrove forests. Nevertheless, Wolanski (2006) has noted that mangroves probably cannot protect the coast against a tsunami wave greater than a threshold level based on some evidence

from observations of the Indian Ocean tsunami. For tsunami being always present a threat to lives and property along the most coasts of the world, it remains an important for estimating the effectiveness of the coastal vegetation on the tsunami impact.



Many numerical and experimental approaches have been developed in recent years to help understanding the tsunami wave interactions with coastal vegetation. The coastal vegetation was idealized by a group of rigid cylinders in most investigations. Huang et al. (2011) performed both experiments and a numerical model by considering solitary wave propagating on emergent rigid vegetation and found that dense vegetation may reduce the wave transmission because of the

increased wave reflection and energy dissipation into turbulence in vegetation. By using both direct simulation and macroscopic approach, Maza et al. (2015) simulated numerically the interaction of solitary waves with emergent rigid cylinders based on the arrangement of laboratory experiments of Huang et al. (2011). Previous approaches (e.g. Anderson et al., 2011; Huang et al., 2011; Maza et al., 2015; Wu et al., 2016) assumed that the idealized mangrove vegetation is stationary and neglected the plant motion with the wave. In the reality, however, vegetation stems may move like cantilever

or whip driven by waves (Paul et al. 2012).

For most of natural vegetation is deformable, which reduces flow resistance, thus it is robust way to be not neglecting vegetation motion. Accordingly, this study tries to provide a better physical model in the numerical simulation by considering vegetation motion coupled with waves to investigate wave damping performance by mangroves. A direct numerical model based on computational fluid dynamics (CFD) is adopted in this paper for simulating the wave damping

characteristics including both the stationary and moving vegetation.

## 2 Numerical model description

Among a number of open source CFD codes available, IHFORM (Higuera et al., 2013; 2014) is specially designed for coastal engineering applications. IHFORM model was used in Maza et al. (2015) for direct simulation of the solitary wave interacting with the stationary vegetation. Alternatively, the model Flow-3D (Flow Science, Inc., 2012) is applied in this

paper to conduct the numerical simulations to involve the motion of the vegetation accompanied by wave. Flow-3D provides exclusively the FAVOR (fractional area/volumes obstacle representation) technique (Hirt, 1993) and the general moving object (GMO) model that is capable of simulating the rigid body motion dynamically coupled with fluid flow. The FAVOR technique retains rectangular elements with a simple Cartesian grid system and shown to be one of the most efficient methods to treat the immersed solid bodies (Xiao, 1999). The free water surface tracking in the model is accomplished by

using volume of fluid (VOF) method (Hirt and Nichols, 1981).

Referring to previous literature, the problem is idealized by solitary wave passing on a group of emergent rigid cylinders. Considering the fluid to be incompressible, the continuity and momentum equations for a moving object formulated with area and volume fraction functions are given as

$$\frac{\partial(u_i A_i)}{\partial x_i} = -\frac{\partial V_F}{\partial t},\tag{1}$$

$$\frac{\partial u_i}{\partial t} + \frac{1}{V_F} A_j u_j \frac{\partial u_i}{\partial x_j} = -\frac{1}{\rho} \frac{\partial p}{\partial x_i} + g_i + \frac{1}{\rho V_F} \frac{\partial}{\partial x_j} \left[ A_j[(\mu + \rho v_t) S_{ij} - \frac{2}{3} \rho k \delta_{ij})] \right]\tag{2}$$





where $V_F$ is the fractional volume open to the flow and $A$ is the fraction area for the subscript direction, $S_{ij} = (\partial u_i / \partial x_j + \partial u_j / \partial x_i)$, $u_i$ is the mean velocity component in the $i$th direction, the subscripts $= 1, 2, 3$ represent $x$-, $y$- and $z$-directions, respectively, $p$ is the pressure intensity, $\rho$ is the fluid density, $g$ is the gravitational acceleration, $\mu$ is the absolute viscosity, $v_t$ is the eddy viscosity, $k$ is the turbulent kinetic energy, and $\delta_{ij}$ is the Kronecker delta function such that $\delta_{ij} = 1$

when $i = j$; $\delta_{ij} = 0$, when $i \neq j$. The eddy viscosity $v_t$ is related to the effect of the space and time distribution of the turbulent motion, which is solved here by using the renormalization group method (RNG $k$-$\varepsilon$ model) proposed in Yokhot and Orszag (1986). It is noted that the above governing equations are rendered to standard RANS equations as both $V_F$ and $A$ are set to unity.

  Comparing with the continuity equation for stationary obstacle problems, $-\partial V_F / \partial t$ in equation (1) is equivalent to an

additional volume source term and exists only in mesh cells around the moving object boundary. It can be calculated using

$$-\frac{\partial V_F}{\partial t} = \frac{S_{obj}}{V_{cell}} \vec{V}_{obj} \cdot \vec{n} \tag{3}$$

where $V_{cell}$ is volume of a mesh cell, $S_{obj}$, $\vec{n}$ and $\vec{V}_{obj}$ are respectively surface area, unit normal vector and velocity of the moving object in the mesh cell. The relative transport equation for the VOF function $F$ is given using

$$\frac{\partial F}{\partial t} + \frac{1}{V_F} \frac{\partial (F u_i A_i)}{\partial x_i} = -\frac{F}{V_F} \frac{\partial V_F}{\partial t} \tag{4}$$

According to kinematics, general motion of a rigid body can be divided into a translational motion and a rotational motion. Equations of motion governing the two separate motions are

$$\vec{F}_G = m \frac{d\vec{V}_G}{dt} \tag{5}$$

and

$$\vec{T}_G = [J] \cdot \frac{d\vec{\omega}}{dt} + \vec{\omega} \times ([J] \cdot \vec{\omega}) \tag{6}$$

respectively, where $\vec{F}_G$ is the total force, $m$ is rigid body mass, $\vec{V}_G$ is the mass center velocity, $\vec{T}_G$ is the torque about the object mass centre, $[J]$ is the moment of inertia tensor, and $\vec{\omega}$ is the angular velocity of the rigid body. If the cylinder is considered to swing in the $x$-direction accompanied by wave, angular velocity of the rotating cylinder has one non-zero component only. Then the equations of motion of the cylinder are rendered as

$$T = J\dot{\omega} \tag{7}$$





where $T$, $J$, and $\dot{\omega}$ are total torque, moment of inertia and angular acceleration about the fixed axis. And the velocity of any point P on the rotating cylinder is calculated by $V_P = \omega r_{P/C}$, where $r_{P/C}$ denotes distance from the fixed end C of the cylinder to point P.

In computing the coupling of fluid and rigid body interaction, the velocity and pressure of fluid flow are first solved. The hydrodynamics forces on the rigid body are then obtained and used to calculate the velocity of the rigid body. Then the volume and area fractions are updated according to the new position of the rigid body, and the source term can be calculated using equation (3). The flow field is computed repeatedly until the convergence is achieved.

As for the boundary conditions, the normal stress is in equilibrium with the atmospheric pressure while shearing stress is zero on the free surface. All of the solid surfaces were treated using the no-slip boundary condition. The variation of the turbulent energy and the turbulent energy dissipation on the free surface boundary was set as zero in the normal direction. The solution of solitary wave derived from Boussinesq equations was employed as the incident wave.

## 3 Validation

Huang *et al.* (2011) conducted laboratory experiments in a wave flume for the solitary waves interacting with emergent, rigid vegetation. The vegetation was considered as a group of cylinders which were made of Perpex tubes with a uniform outer diameter of 0.01 m. The computations used the same geometric configuration of Huang's laboratory works. The water depth was uniform and equal to $h = 0.15$ m, and the cylinder height was 0.24 m. The arrangement of cylinders with density of 560 cylinders/m$^2$ and with field length 1.635 m, as shown in Fig. 1, was selected to validate the numerical simulation. There were totally 53 rows of cylinders installed in the numerical tank. The numerical tank was set by 6 m long, 0.55 m wide and 0.3 m height. Note that the verification of the model performance is only implemented by the case of stationary vegetation because of the lack of the experimental data for the moving vegetation.

Two different uniform computational meshes around the cylinder field, 0.002 m and 0.001 m respectively, were used to test the numerical accuracy and the sensitivity to grid size. Fig. 2 shows that FAVOR technique resolved successfully the geometry of cylinders using these two computational grids constructed. Fig. 3 shows the comparison of free surface evolution between the present numerical results and experimental measurements for an incident wave height $H_i = 0.05$ m. The results obtained by the direct simulation using IHFORM in Maza *et al.* (2015) were also shown in the figure. Fig. 4 shows the maximum wave height at each wave gauge probe between the numerical results and experimental measurements. The results depict that the present numerical simulations are in a very good agreement with the laboratory experiments. The results also show almost no differences between both computational meshes used.

## 4 Results and discussion

The above comparisons demonstrated the present numerical model is capable of simulating accurately the wave evolution by the group cylinders. The following simulations are performed for solitary wave passing through both the stationary and



moving cylinders. The surface elevation evolution, flow field variation, and the turbulent kinetic energy dissipation are analysed and compared between these two situations of cylinders. The numerical domain and the arrangement of cylinders used in the following simulations are the same previous section. The water depth was uniform and equal to $h = 0.15$ m, and the cylinder height was 0.24 m. The density of the arrangement is equal to 560 cylinders/m² and the cylinder field length is

1.635 m. The computational mesh with 0.002 m is used for saving computing time based on its achievable accuracy.

The rotating cylinders induced by waves are set-up by the general moving object (GMO) model stated above for coupling the cylinder's motion and fluid flow dynamically. Each cylinder end was attached a torsion spring on the bottom in the model. The spring constant is 1 kgw/m. The specific gravity of the cylinder is set by 0.25. The use of torsion spring could not completely reproduce the natural bending behaviour of the mangrove tree, but it allows the cylinders to swing with the

passing wave to respond the moving effect of the rigid body on the flow field.

**4.1 Flow field evolution**

Fig. 5 shows the snapshots of velocity distribution at the centre line of the tank for the moving and stationary cylinders respectively as the solitary wave crest passing through gauges G3 to G6 for an incident wave height $H_i = 0.05$ m. For the stationary cylinders, it can be observed that the water velocity reduces rapidly as wave crest impinging on the array of

cylinders. The moving cylinder has angular rotation and immerses in the water while the wave crest passing over, thus resulting less water velocity reduction than the stationary cylinders. Fig. 6 shows the comparison of the horizontal velocity profile as wave crest passing through each gauge for the moving and stationary cylinders, which confirms that the motion effect of the cylinder leads to less water velocity reduction.

**4.2 Surface elevation evolution**

The wave surface evolutions for the moving and stationary cylinders respectively are shown in Fig. 7. Before the solitary wave reaches the cylinders, the wave height almost keeps with the same, and the surface evolution has no much difference between the stationary and moving cylinders. However, surface evolution appears difference while and even after the wave propagation over the stationary and moving cylinders. The weakly wave reflection can be found at the front row of the stationary cylinders while it is not obviously for the moving cylinders. The result of free surface evolution depicts that the

stationary cylinders are working better than the moving cylinders for the wave height damping as the waves pass through the cylinders.

Fig. 8 shows the comparison of wave height evolution for the moving and stationary cylinders using different incident wave heights ($H_i = 0.025$ m, 0.04 m, 0.05 m); their relative wave heights are equal to $H_i/h = 0.17$, 0.25 and 0.33, respectively. It can be seen that the wave height ratio ($H/H_i$) decays rapidly for the stationary cylinders but mildly for the moving

cylinders.

As shown in Fig. 9, the maximum wave height damping ratio for the moving cylinders is approximately 26% yet it is nearly 61% for the stationary cylinders after the wave passed through the cylinders. Here the wave height damping ratio is





defined by $H_D = (H - H_i)/H_i$, in which $H$ is the local wave height and $H_i$ is the incident wave height. The simulated results demonstrate that the wave height decay might be overestimated if the mangrove forests were idealized as stationary state for evaluating their effectiveness on the tsunami damping.

### 4.3 Turbulent kinetic energy dissipation evolution

The turbulent kinetic energy will be generated and dissipated during the wave interacting with the group of cylinders. The turbulent kinetic energy dissipation is the important mechanism of the wave height damping. The turbulent kinetic energy ($k$) and dissipation of the turbulent kinetic energy ($\varepsilon$) are obtained from the RNG $k$-$\varepsilon$ turbulent closure model while solving general RANS equations.

    Fig. 10 shows snapshots of the distribution of the turbulent kinetic energy dissipation (DTKE) along the cylinder array for
the moving and stationary situations respectively when the wave crest passing over gauges G3 to G6 for an incident wave height $H_i = 0.05$ m. It can be seen that the turbulent kinetic energy dissipation happens as the wave crest impinging on the cylinders. For stationary cylinders, the largest turbulent energy dissipation occurs in the region between gauges G3 and G4 as the wave crest passing through the gauge G4. But for the moving cylinders, the largest turbulent energy dissipation occurs as the wave crest passing through the gauge G5 and the dissipation is less than the stationary cylinders case.

The total DTKE shown in Fig. 11 is calculated from the total computed meshes of the numerical tank at the time when wave crest passing each gauge. It shows that the total DTKE increases rapidly to a maximum value after wave crest passed the front cylinders. After having its maximum value at gauge G4 for the stationary cylinders and gauge G5 for the moving cylinders, the total DTKE decreased. It is reasonable to see that the larger incident wave height has the larger DTKE. It is also observed that the stationary cylinders have larger DTKE before the wave passing gauge 5 but the moving cylinders have
larger DTKE after the wave passed gauge G5. This is because that a little turbulent kinetic energy could be generated and dissipated by moving cylinders during the restoring process to the stand position after the wave passed over. Nevertheless, the integral DTKE for the moving cylinders is less than that of the stationary cylinders.

### 5. Conclusions

    A numerical simulation based on the general RANS equations and RNG $k$-$\varepsilon$ turbulent model was implemented to
investigate the moving effect of coastal vegetation on the damping of tsunami wave. The vegetation was idealized by a group of emergent, rigid cylinders. The FAVOR technique and moving object (GMO) model provided in Flow-3D code were employed in this paper for simulating the coupling of fluid and rigid body interaction. The evolutions of wave height, flow field and turbulent kinetic energy dissipation rate for both stationary and moving cylinder are investigated. Due to the moving effect of the cylinders accompanied by the wave, the numerical results showed that their wave height damping and
the turbulent kinetic energy dissipation rate were markedly less than those of stationary cylinders. This result leads to an



important note that it might be overestimated for tsunami damping if the coastal vegetation is represented as a stationary state.

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



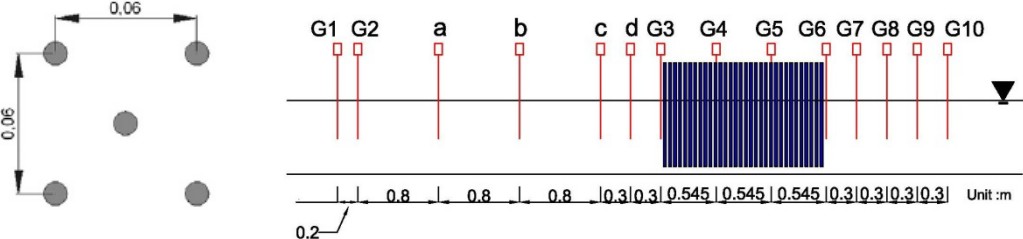

**Figure 1.** Cylinder cell arrangement (left), field length (right) and locations of wave probes for the computations.

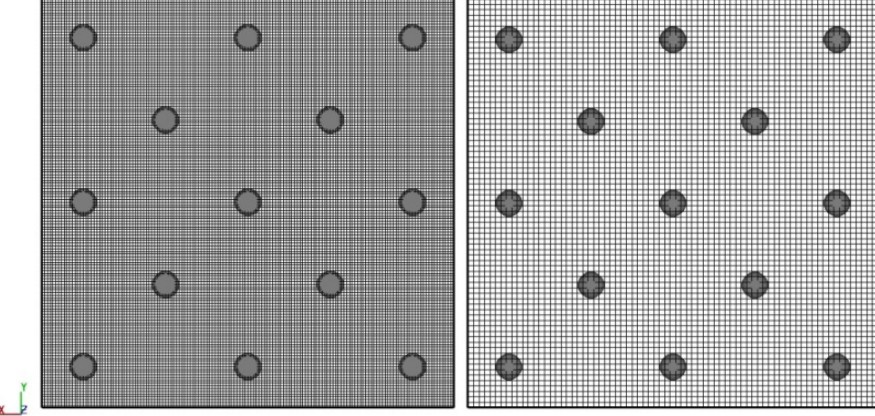

**Figure 2.** FAVORized geometry of cylinders and constructed computational rectangular grid with 0.001 m (left) and 0.002 m (right) (Note: only four cylinder cells are depicted).





**Figure 3.** Comparison of free surface evolution between numerical and experimental results for $H_i$ = 0.05 m.





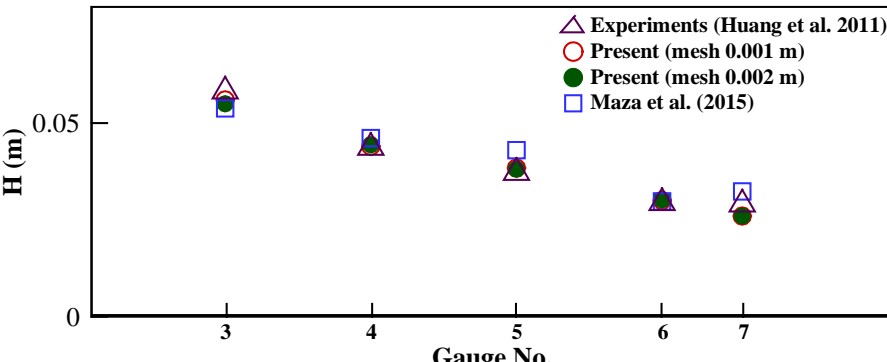

**Figure 4.** Comparison between the numerical results and experimental measurements for the wave height at each gauge for $H_i$ = 0.05 m.

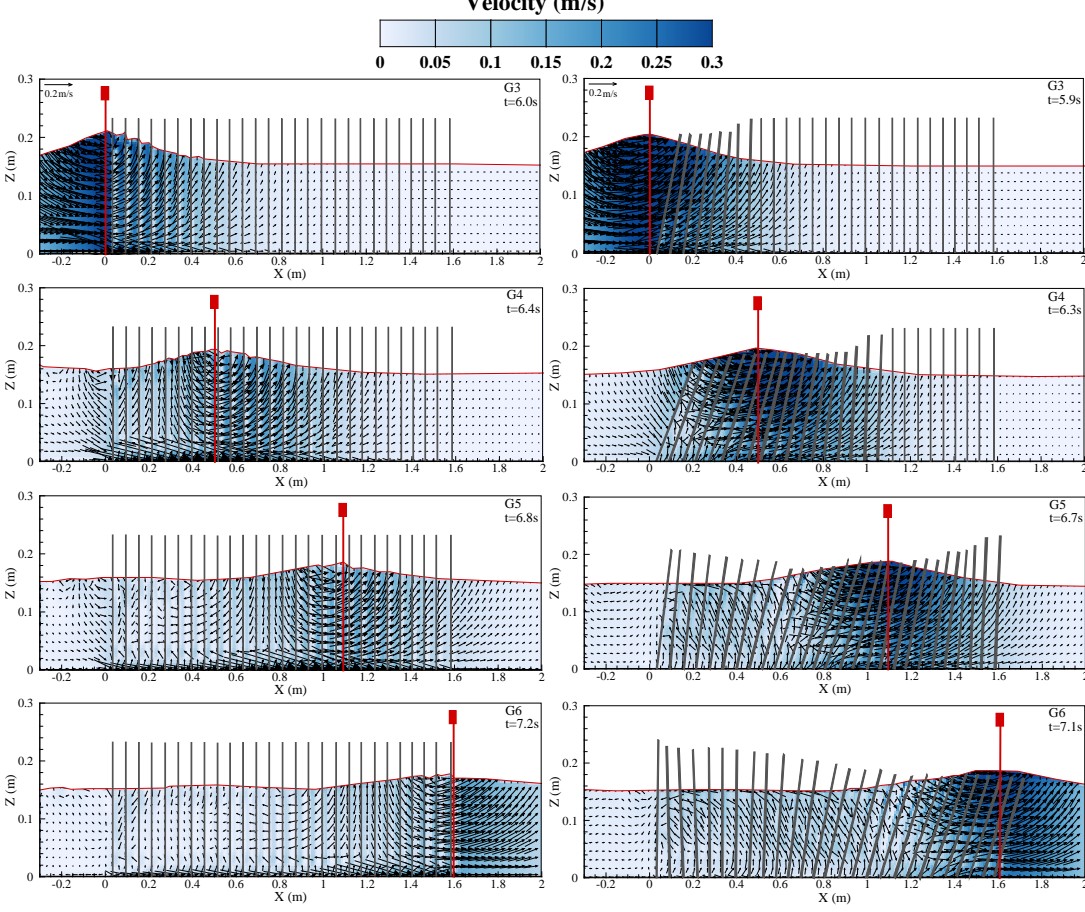

10   **Figure 5.** The snapshots of the velocity distribution for the stationary cylinders (left) and moving cylinders (right) for $H_i$ = 0.05 m.


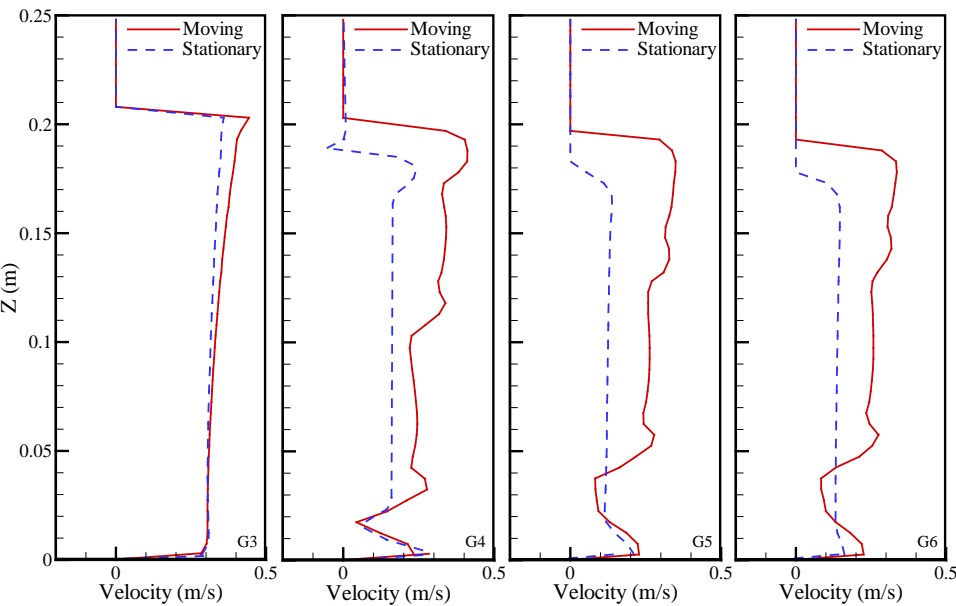

**Figure 6.** Comparison of the horizontal velocity profile for the moving and stationary cylinders as wave crest passing through each gauge for $H_i$ = 0.05 m.

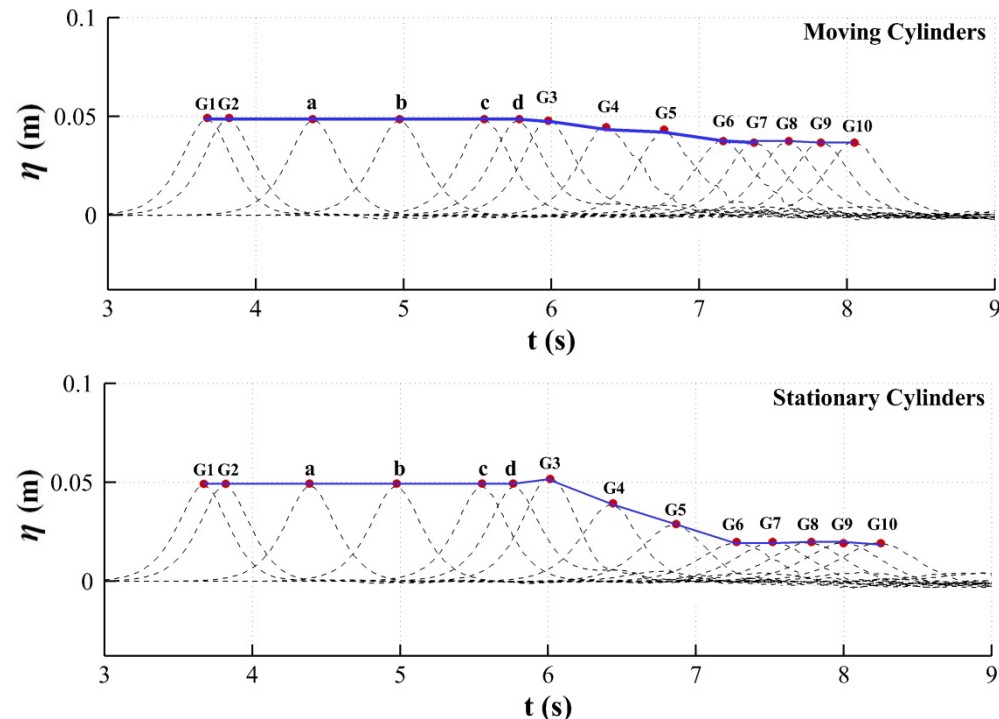

**Figure 7**. Wave surface evolutions of the moving and stationary cylinders for $H_i$ = 0.05 m.



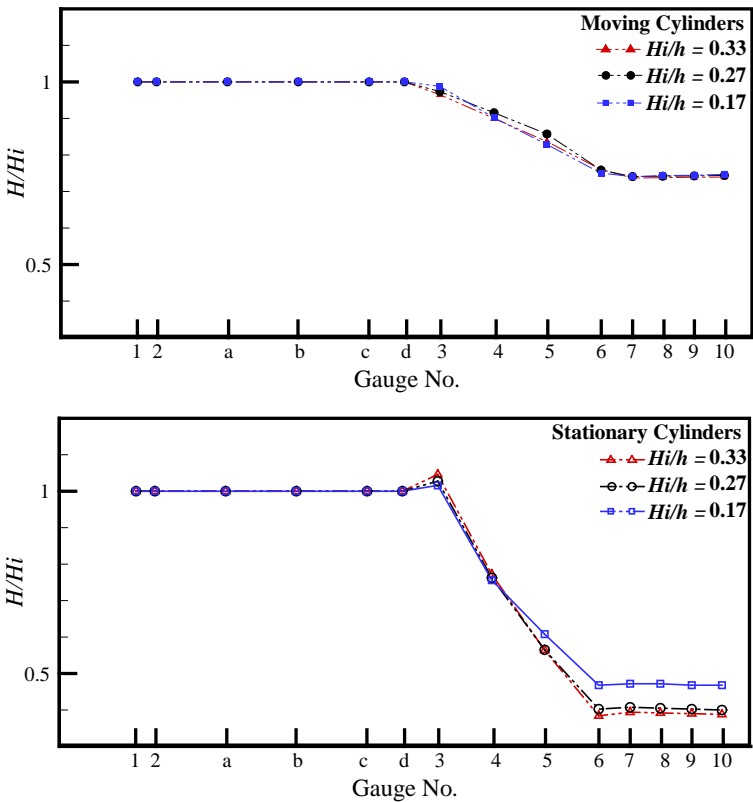

**Figure 8.** Wave height evolutions for moving and stationary cylinders of different relative wave heights ($H_i/h$).

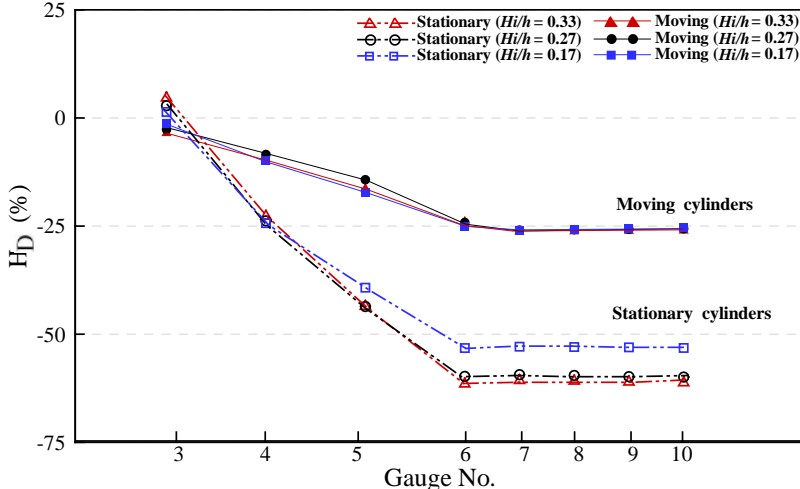

5  **Figure 9.** Comparison of wave height damping ratio between moving and stationary cylinders for different incident wave heights.





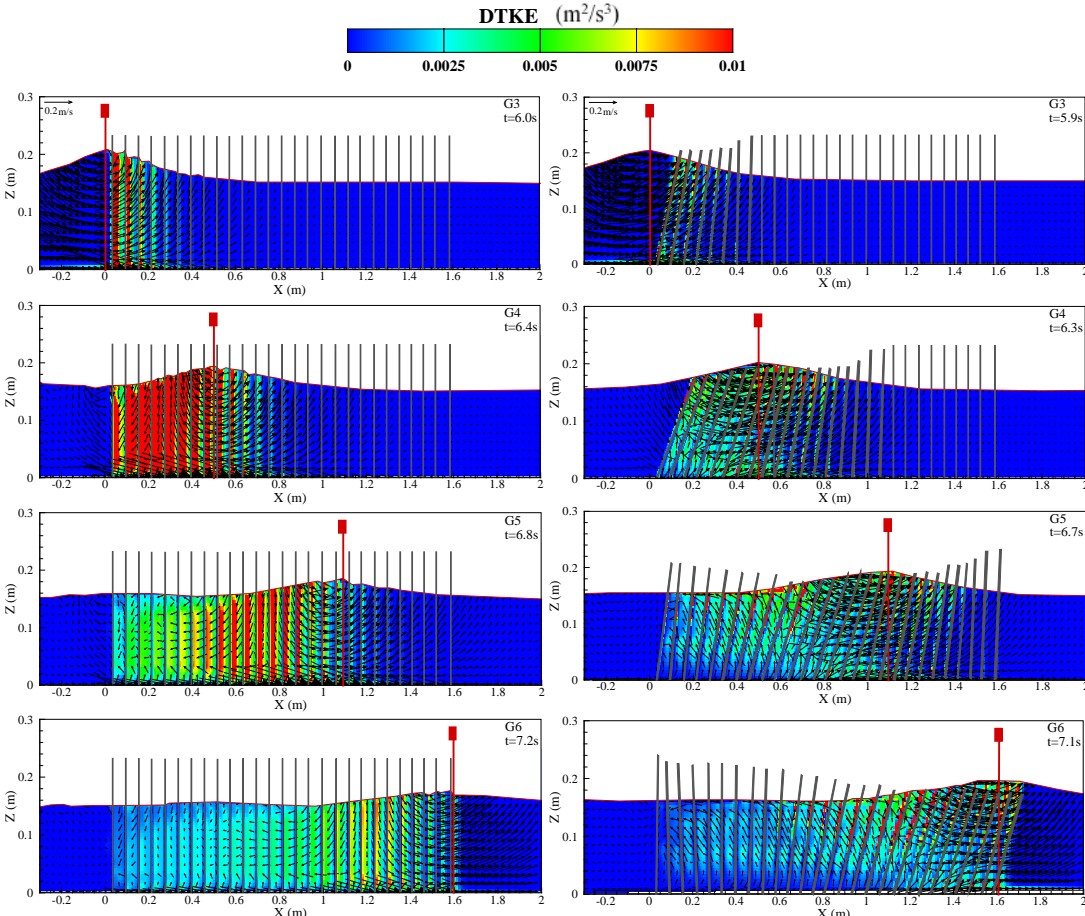

**Figure 10.** Distribution of turbulent kinetic energy dissipation (DTKE) for stationary cylinders (left) and moving cylinders (right) for $H_i$ = 0.05 m.



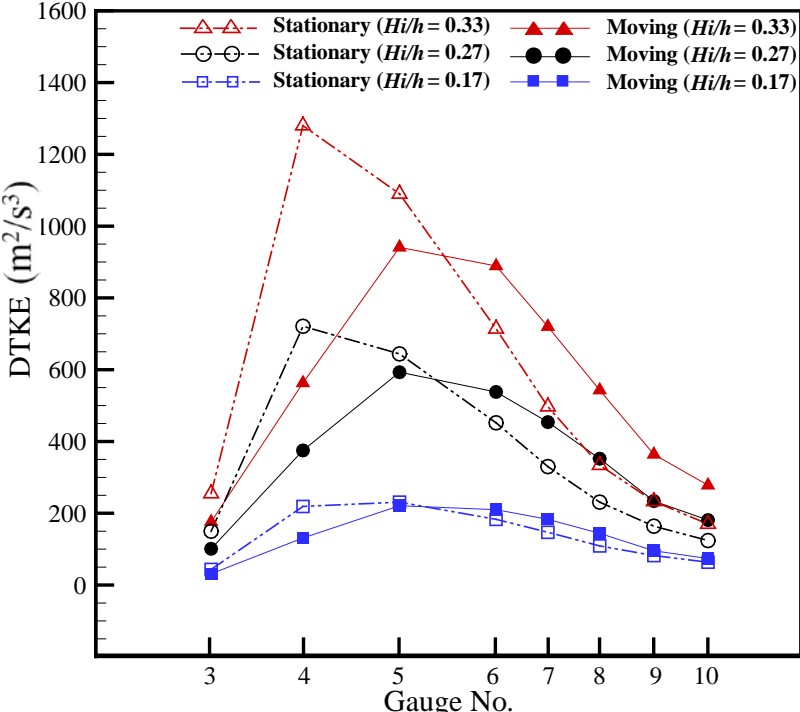

**Figure 11.** Comparison of total DTKE for moving and stationary cylinders for different incident wave heights when wave crest passing through each gauge.