# Peer review of "Simulations of Moving Effect of Coastal Vegetation on Tsunami Damping"

_Natural Hazards and Earth System Sciences, 2016_

## Short Comment (SC1) · 10 Jan 2017

General comments:

The discussion paper "Simulations of moving effect of coastal vegetation on tsunami damping" is well structure and covers an interesting topic, i.e. the effect of vegetation movement under flow action in the resulting wave attenuation. However, there are some weak points in the statement of the problem as well as along the validation and discussion of the results. These points are highlighted in the following sections. In addition, a strong effort should be done in the English grammar correction since there are some grammar mistakes and sentences that are not well written.

Specific comments:

[Figure]

1. The manuscript is focus on mangrove forests. However, authors do not provide with evidences of mangroves movement under wave action. Previous studies have argued that mangroves are stiff enough to be considered as rigid-not moving bodies under flow action (e.g.: Zhang et al. 2015, "Hydrodynamics in mangrove prop roots and their physical properties"). Since the model used in the manuscript only considers the movement with respect to the bottom by using a spring, authors maybe can state this is a very simplified way to represent some movements induced by sediment scour and mangrove uproot states, as shown in the field campaing performed by Yanagisawa et al. (2009): "The reduction effects of mangrove forest on a tsunami based on field surveys at Pakarang Cape, Thailand and numerical analysis".

2. Introduction: in general literature review of the problem is poor. a. There are more recent papers about tsunami reduction by mangroves than the ones highlighted in the manuscript. b. Another important point is the lack of literature review on models able to capture vegetation flexibility under waves action (e.g.: Maza et al. 2013, "A coupled model of submerged vegetation under oscillatory flow using Navier–Stokes equations"). Since they are proposing a new model to represent vegetation motion they should perform a literature review of this issue to find the models that have been already proposed to solve that problem. This will allow them to highlight the advantages of the proposed approach. c. Authors mentioned Paul et al. (2012) work to point out the importance of considering vegetation motion but that paper was performed for submerged vegetation under tidal current action, something that is far from the problem faced here (emergent vegetation under solitary wave action). They should find a different reference to point out the importance of considering that aspect in mangrove forests or explain the implications of Paul's paper in their work. d. Also, in the last paragraph they talk about "vegetation is deformable" but mangrove, in general are not.

3. Numerical model description: a. There are some variables that are not defined: "uj, xi, xj, t". Notation is not consistent along the different equations: authors used sometimes vector notation, in some other equations they use Einstein notation. You

should write everything following the same criterion. b. Turbulence closure model: authors should explain why they are using RNG k-epsilon model instead some other options such as k-omega SST that is aim to be the most suitable one for cases where there is flow-structure (in this case cylinder) interaction.

4. Validation: a. Different mesh discretizations can be better understand if authors provide with the number of points defining cylinder's diameter for each case. In addition, figure 2 does not provide with very valuable information, a zoom in around one cylinder will be useful to better visualize the mesh. b. No validation is shown for cases where cylinders move. Experiments consider here where performed using rigid stationary cylinders so that information is not available. However, authors can refer some other applications of the GMO model to shows its capabilities on solving similar problems. c. Figure 4: y-axis scale is not providing enough information; there are only two points. Furthermore, it will be more helpful to set the axis as no dimensional variables: H/Hincident and X/Lcylinders, for example. Data from Maza et al. (2015) looks different than the one provided at panel C in figure 14 of that paper. How is that data obtained?

5. Results and discussion: a. Authors are using a spring constant equal to 1kgw/m and specific cylinder gravity equal to 0.25, why? They should explain where these values come from. b. Section 4.2: "The weakly wave reflection can be found at the front row of the stationary cylinders while it is not obviously for the moving cylinders" how do you see this effect? I do not see any significant reflected wave for any case. c. Figure 8 is not giving any valuable additional information. d. Figure 10 shows turbulent kinetic energy dissipation or turbulent kinetic energy (k)? e. Section 4.3: "DTKE is calculated from the total computed meshes of the numerical tank", do you mean it is the integrated value in the entered domain? It is not clear how you compute this value. In addition, authors say they are calculating DTKE after wave crest passes each gauge, is that a good way to evaluate TKE evolution in the problem? The maximum TKE is produced after wave crest passes, that is there is a lag between the maximum wave elevation and the occurrence of maximum TKE. Then, I think values represented in figure 11 are

not providing with all the required information to understand what is happening with this variable, especially when thinking about maximum TKE values. Instead, authors can provide, for example, with the maximum TKE value at each mesh cell recorded in the whole simulation. That way they will provide with a map of the maximum TKE along the entered domain for both cases (stationary and moving).

6. Conclusions: authors talk about TKE dissipation rate but they are not providing with any rate values.

Technical corrections:

1. IHFOAM is misspelled along the entered manuscript; authors are using IHFORM, which is wrong.

2. Page 2, line 11: change "For most of" to "Most of".

3. Page 2, line 20: rephrase "to involve the motion of the vegetation accompanied by wave", by something like "including vegetation motion under waves action".

4. Page 2, line 23: "shown" by "has been shown".

5. Page 4, validation section: there are several grammar mistakes or sentences that are not very well written such as: "The arrangement of cylinders with density of 560 and with field length...", "Fig. 4 shows the maximum wave height at each wave gauge probe between numerical results and experimental...".

6. Figure 5: it would be better if the color scale is different (similar to the one shown in figure 10 for example) to better observe the differences between two approaches.

7. Figure 6: Wave gauges names are very small, please increase the names or set them on top of the panels.

8. Page 5, line 3: "the same previous section" to "the same as in previous section".

9. Page 5, lines 3 – 5: you have already explain the experimental set-up so you don't

need to include again the values of the water depth and cylinders field.

10. Section 4: there are many grammar mistakes: "while the wave crest passing over...", "resulting less water velocity...", "keeps with the same", "yet it is nearly", "the important mechanism", "the dissipation is less than the stationary".....

11. Page 7, line 1: "note that it might be overestimated for tsunami damping if..." to "that is, tsunami damping can be overestimated if...".
* * *

---

## Referee Comment (RC2) · Anonymous Referee #2 · 2 Feb 2017

General comments:

A numerical investigation of tsunami damping over coastal vegetation is presented. The authors show that the moving effects of the emergent cylinders have less wave height damping and turbulent kinetic energy dissipation than the stationary, emergent cylinders. The paper is well structured and the purpose of the paper is clear; however, the English grammar and sentences are not well written and need to be improved. The manuscript can be acceptable in Natural Hazards and Earth System Sciences after a major revision with further treatment of the following sections:

Specific comments:

Page 2, line 9 – 10: Paul et al., 2012 showed submerged seagrass mimics attenuate wave height. , but authors need to provide more literature reviews to convince readers

mangroves are considered as moving vegetation motion.

Page 4, line 19 – 20: There are several references referring to flexible vegetation and authors need to provide in the introduction and to validate your model.

Page 5, line 3 – 4: The paragraph is repeated as before. They should be deleted.

Page 5, line 5: Figure 2 is not clear to show the meshes around the cylinders and please explain why the mesh with 0.002 m can achieve accuracy.

Page 5, line 7 – 8: Authors need to explain why you used the specific of the cylinder 0.25 and the spring constant 1 kgw/m and why they are appropriate coefficients to the moving mangroves.

Page 5, line 23: It would be nice to calculate the wave reflection coefficient from the stationary cylinders and the moving cylinders.

Page 5, line 31 – 32: usually we use Dalrymple et al. (1984) equation or Kobayashi et al. (1993) equation to calibrate the wave height damping coefficients according to your Fig. 8 from G3 to G6 for different incident wave heights.

Page 6, line 7 – 8: Can you explain why you used RNG k-epsilon model instead of k-epsilon or k-omega model?

Page 6, line 7: Authors mentioned TKE dissipation rate (epsilon) but did not discuss that. It would be interesting to see the vertical profiles of TKE budget for both stationary and moving cases.

Page 6, line 15: Please define DTKE. How did you calculate DTKE values in your model?

Page 9: Figure 1., the figure doesn't look good. Some number is not necessary to show from G2 – G3 and the fonts are too small.

Page 11: Figure 4., it seemed like Maza et al. (2015) results at G7 have wave reflection.

Can you explain why?

Page 13: Figure 8., y axis would plot only vegetation field, e.g. G3 to G6 and you can define Kv as the vegetation transmission, Kv(x)

Technical corrections:

1. Page 2, line 17: Typo. Substitute 'IHFORM' with 'IHFOAM.'

2. Page 4, line 25: Typo. Substitute 'IHFORM' with 'IHFOAM.'

3. Page 5, line 17: 'through each cylinder' should be re-written as 'through G3 to G6' and corresponding Fig. 6 needed to increase the font size of G3 to G6. Also, caption needs to be changed as well.

4. Page 6, line 26: Typo. Substitute 'moving object (GMO)' with 'general moving object (GMO).'

---

## Author Comment (AC1) · 28 Feb 2017

The authors would like to express their deep appreciation to Professor Maza for the invaluable comments and suggestions, which have led us to improve the quality of the paper. Detailed responses are listed below.

General comments:

The discussion paper "Simulations of moving effect of coastal vegetation on tsunami damping" is well structure and covers an interesting topic, i.e. the effect of vegetation movement under flow action in the resulting wave attenuation. However, there are some weak points in the statement of the problem as well as along the validation and discussion of the results. These points are highlighted in the following sections. In addition, a strong effort should be done in the English grammar correction since there

are some grammar mistakes and sentences that are not well written. Response: The authors have followed the reviewer's suggestions to make the necessary revisions, which can be seen as RED fonts in the revised manuscript. We also change "Moving Effect" in the title to "Swaying Effect" for complying with the topic of this paper.

Specific comments:

1. The manuscript is focus on mangrove forests. However, authors do not provide with evidences of mangroves movement under wave action. Previous studies have argued that mangroves are stiff enough to be considered as rigid-not moving bodies under flow action (e.g.: Zhang et al. 2015, "Hydrodynamics in mangrove prop roots and their physical properties"). Since the model used in the manuscript only considers the movement with respect to the bottom by using a spring, authors maybe can state this is a very simplified way to represent some movements induced by sediment scour and mangrove uproot states, as shown in the field campaing performed by Yanagisawa et al. (2009): "The reduction effects of mangrove forest on a tsunami based on field surveys at Pakarang Cape, Thailand and numerical analysis".

Response: We have added several literatures about the mangroves movement under wave action, which can be seen as RED fonts in Introduction. We also referred to Kazemi et al. (2015) to demonstrate that the flexibility of the roots was modeled by attaching rigid cylinders to torsional connectors. We also state that this is also a simplified way to represent some movements of mangroves induced by sediment scour, tilting or uprooting states (page 2, Lines 24-28).

2. Introduction: in general literature review of the problem is poor. a. There are more recent papers about tsunami reduction by mangroves than the ones highlighted in the manuscript. b. Another important point is the lack of literature review on models able to capture vegetation flexibility under waves action (e.g.: Maza et al. 2013, "A coupled model of submerged vegetation under oscillatory flow using Navier–Stokes equations"). Since they are proposing a new model to represent vegetation motion

they should perform a literature review of this issue to find the models that have been already proposed to solve that problem. This will allow them to highlight the advantages of the proposed approach. c. Authors mentioned Paul et al. (2012) work to point out the importance of considering vegetation motion but that paper was performed for submerged vegetation under tidal current action, something that is far from the problem faced here (emergent vegetation under solitary wave action). They should find a different reference to point out the importance of considering that aspect in mangrove forests or explain the implications of Paul's paper in their work. d. Also, in the last paragraph they talk about "vegetation is deformable" but mangrove, in general are not.

Response: a. We have added more than10 references in general literature review. b. The paper of Maza et al. (2013) has been referred (page 2, Lines 22-24). c. The reference of Paul et al. (2012) has been removed. d. The mistake has been revised.

3. Numerical model description: a. There are some variables that are not defined: "uj, xi, xj, t". Notation is not consistent along the different equations: authors used sometimes vector notation, in some other equations they use Einstein notation. You should write everything following the same criterion. b. Turbulence closure model: authors should explain why they are using RNG k-epsilon model instead some other options such as k-omega SST that is aim to be the most suitable one for cases where there is flow-structure (in this case cylinder) interaction.

Response: a. Those variables have been defined. All equations have been expressed in tensor forms. b. The use of RNG k-epsilon model was described in pages 3 and 4. When the RANS equations with RNG model was solved by FAVOR technique and VOF method, we can obtain accurate results for the present wave-structure interaction. Based on the comparisons shown in Figs. 3 and 4, we can find that the accuracy between SST k-omega used in Maza et al (2015) and the present RNG is almost no difference.

4. Validation: a. Different mesh discretizations can be better understand if authors

provide with the number of points defining cylinder's diameter for each case. In addition, figure 2 does not provide with very valuable information, a zoom in around one cylinder will be useful to better visualize the mesh. b. No validation is shown for cases where cylinders move. Experiments consider here where performed using rigid stationary cylinders so that information is not available. However, authors can refer some other applications of the GMO model to shows its capabilities on solving similar problems. c. Figure 4: y-axis scale is not providing enough information; there are only two points. Furthermore, it will be more helpful to set the axis as no dimensional variables: H/Hincident and X/Lcylinders, for example. Data from Maza et al. (2015) looks different than the one provided at panel C in figure 14 of that paper. How is that data obtained?

Response: a. The number of points defining cylinder's diameter for each mesh has been added. In addition, figure 2 has been represented using one cylinder. (page 5, Lines 29-30 and page 11, Fig. 2) b. Unfortunately, the experimental information on the swaying cylinders by solitary waves is lacking. So we could only implement the validation by the case of stationary cylinders. We have added some references to state other applications of the GMO model to shows its capabilities on solving similar problems. (Page 5, Lines 10-12) c. Figure 4 has been revised by using H/Hi vs x/L. We would like to thanks Prof. Maza providing their data for our use for Figs. 3 and 4.

5. Results and discussion: a. Authors are using a spring constant equal to 1kgw/m and specific cylinder gravity equal to 0.25, why? They should explain where these values come from. b. Section 4.2: "The weakly wave reflection can be found at the front row of the stationary cylinders while it is not obviously for the moving cylinders" how do you see this effect? I do not see any significant reflected wave for any case. c. Figure 8 is not giving any valuable additional information. d. Figure 10 shows turbulent kinetic energy dissipation or turbulent kinetic energy (k)? e. Section 4.3: "DTKE is calculated from the total computed meshes of the numerical tank", do you mean it is the integrated value in the entered domain? It is not clear how you compute this value. In addition, authors say they are calculating DTKE after wave crest passes each gauge, is that a

good way to evaluate TKE evolution in the problem? The maximum TKE is produced after wave crest passes, that is there is a lag between the maximum wave elevation and the occurrence of maximum TKE. Then, I think values represented in figure 11 are not providing with all the required information to understand what is happening with this variable, especially when thinking about maximum TKE values. Instead, authors can provide, for example, with the maximum TKE value at each mesh cell recorded in the whole simulation. That way they will provide with a map of the maximum TKE along the entered domain for both cases (stationary and moving).

Response: a. We have stated the use of spring constant and specific cylinder gravity, as in page 6, Lines 18-23. Besides, we added Figs. 6 and 7 to discuss the wave height evolution and the deflection angle variation of cylinder using different spring constants. b. The statement about wave reflection has been omitted because it is not significant. c. The original Fig. 8 has been removed. d. Original Fig. 10 shows turbulent kinetic energy dissipation (epsilon). In the revised version, we show snapshots of both turbulent kinetic energy (k) and turbulent kinetic energy dissipation (epsilon), as shown in Figs. 11 and 12. e. We have rewritten the section 4.3 for turbulent kinetic energy (TKE) evolution. The time variations of total TKE at each section (Fig. 13) is added in the revised version to show the time lag between the occurrence of maximum TKE of the stationary cylinders and the maximum wave elevation. We also added the evolution of vertical profile of TKE to show the swaying effect of cylinders will induce multiple shear layers.

6. Conclusions: authors talk about TKE dissipation rate but they are not providing with any rate values. Response: We have discussed more about the damping of turbulent kinetic energy evolution along the cylinder array in Section 4.3 for the Conclusions.

Technical corrections:

1. IHFOAM is misspelled along the entered manuscript; authors are using IHFORM, which is wrong.

[Figure]

Response: The typo has been corrected.

2.Page 2, line 11: change "For most of" to "Most of".

Response: The original statement has been removed.

3. Page 2, line 20: rephrase "to involve the motion of the vegetation accompanied by wave", by something like "including vegetation motion under waves action".

Response: The statement has been rephrased (page 3, Line 5).

4. Page 2, line 23: "shown" by "has been shown".

Response: The typo has been corrected.

5. Page 4, validation section: there are several grammar mistakes or sentences that are not very well written such as: "The arrangement of cylinders with density of 560 and with field length...", "Fig. 4 shows the maximum wave height at each wave gauge probe between numerical results and experimental...".

Response: The grammar mistakes have been corrected.

6. Figure 5: it would be better if the color scale is different (similar to the one shown in figure 10 for example) to better observe the differences between two approaches.

Response: The color scale of this figure (now is Fig. 9) has been improved.

7. Figure 6: Wave gauges names are very small, please increase the names or set them on top of the panels.

Response: The fonts of this figure and all figures have been enlarged.

8. Page 5, line 3: "the same previous section" to "the same as in previous section".

Response: The mistake has been corrected.

9. Page 5, lines 3 – 5: you have already explain the experimental set-up so you don't need to include again the values of the water depth and cylinders field.
Response: We have removed the repeated description.

10. Section 4: there are many grammar mistakes: "while the wave crest passing over. . .", "resulting less water velocity. . .", "keeps with the same", "yet it is nearly", "the important mechanism", "the dissipation is less than the stationary". . ..

Response: These grammar mistakes have been corrected.

11. Page 7, line 1: "note that it might be overestimated for tsunami damping if. . ." to "that is, tsunami damping can be overestimated if. . .".

Response: We have corrected it following your suggestion.

Please also note the supplement to this comment:
http://www.nat-hazards-earth-syst-sci-discuss.net/nhess-2016-353/nhess-2016-353-AC1-supplement.pdf

---

## Author Comment (AC2) · 28 Feb 2017

The authors would like to express their deep appreciation to Anonymous Reviewer for the invaluable comments and suggestions, which have led us to improve the quality of the paper. Detailed responses are listed below.

General comments:

A numerical investigation of tsunami damping over coastal vegetation is presented. The authors show that the moving effects of the emergent cylinders have less wave height damping and turbulent kinetic energy dissipation than the stationary, emergent cylinders. The paper is well structured and the purpose of the paper is clear; however, the English grammar and sentences are not well written and need to be improved. The manuscript can be acceptable in Natural Hazards and Earth System Sciences after a

major revision with further treatment of the following sections:

Response: The authors have followed the reviewer's suggestions to make the necessary revisions, which can be seen as RED fonts in the revised manuscript. We also change "Moving Effect" in the title to "Swaying Effect" for complying with the topic of this paper.

Specific comments:

1. Page 2, line 9 – 10: Paul et al., 2012 showed submerged seagrass mimics attenuate wave height, but authors need to provide more literature reviews to convince readers mangroves are considered as moving vegetation motion.

Response: The reference of Paul et al. (2012) has been removed. We have added more than10 references about coastal vegetation involving the flexible effect in general literature review.

2. Page 4, line 19 – 20: There are several references referring to flexible vegetation and authors need to provide in the introduction and to validate your model.

Response: There are a lot of literature regarding the flexible vegetation, but most of them concern high stem plants (seagrasses) under flow action. We have added more than 10 references about flexible tree vegetation under wave action in the introduction. Unfortunately, the experimental information on the swaying cylinders under solitary wave action is lacking. So we could only implement the model validation by the case of stationary cylinders.

3. Page 5, line 3 – 4: The paragraph is repeated as before. They should be deleted.

Response: This repeated paragraph has been removed.

4. Page 5, line 5: Figure 2 is not clear to show the meshes around the cylinders and please explain why the mesh with 0.002 m can achieve accuracy.

Response: Figure 2 has been represented using one cylinder which can show clearly

the meshes. We added one figure (Fig. 3) to compare the accuracy with Maza et al. (2015) using meshes of 0.001 m and 0.002 m. It was found that there are almost no differences between the free surface or wave height results using both computational meshes. But we use the fine mesh (0.001m) to simulate the illustration examples for discussion.

5. Page 5, line 7 – 8: Authors need to explain why you used the specific of the cylinder 0.25 and the spring constant 1 kgw/m and why they are appropriate coefficients to the moving mangroves.

Response: We have stated the use of spring constant and specific cylinder gravity, as in page 6, Lines 18-23. Besides, we added Figs. 6 and 7 to discuss the wave height evolution and the deflection angle variation of cylinder for different spring constants.

6. Page 5, line 23: It would be nice to calculate the wave reflection coefficient from the stationary cylinders and the moving cylinders.

Response: The statement about wave reflection has been omitted because it is not significant.

7. Page 5, line 31 – 32: usually we use Dalrymple et al. (1984) equation or Kobayashi et al. (1993) equation to calibrate the wave height damping coefficients according to your Fig. 8 from G3 to G6 for different incident wave heights.

Response: The empirical formulations proposed in Dalrymple et al. (1984) and Kobayashi et al. (1993) are applied for the situation of periodic waves. But the concern of the present paper is solitary wave that their formulas unfortunately could not be applied in this paper.

8. Page 6, line 7 – 8: Can you explain why you used RNG k-epsilon model instead of k-epsilon or k-omega model?

Response: The use of RNG k-epsilon model was described in pages 2 and 3. When the RANS equations with RNG model was solved by FAVOR technique and VOF

method, we can obtain accurate results for the present wave-structure interaction. Based on the comparisons shown in Figs. 3 and 4, we can find that the accuracy between SST k-omega used in Maza et al (2015) and the present RNG is almost no difference.

9. Page 6, line 7: Authors mentioned TKE dissipation rate (epsilon) but did not discuss that. It would be interesting to see the vertical profiles of TKE budget for both stationary and moving cases.

Response: We have rewritten about the section 4.3 about the turbulent kinetic energy. The formulations of the RNG model also has been added in Section 2. The TKE (k) and DTKE (epsilon) are obtained directly from the RNG model that we did not need to compute each component of TKE budget like one-equation model. Instead of TKE budget profile, we have added the vertical profile evolution of TKE to show the swaying effect of cylinders will induce multiple shear layers.

10. Page 6, line 15: Please define DTKE. How did you calculate DTKE values in your model?

Response: The transport equations of TKE (k) and DTKE (epsilon) of the RNG model has been added in Section 2. The values of k and epsilon are obtained directly by solving RANS and RNG equations.

11. Page 9: Figure 1., the figure doesn't look good. Some number is not necessary to show from G2 – G3 and the fonts are too small. Response: The quality of Figure 1 has been improved.

12. Page 11: Figure 4., it seemed like Maza et al. (2015) results at G7 have wave reflection. Can you explain why?

Response: We would like to thanks Prof. Maza providing their data for our use for Figs. 3 and 4. We have corrected the mistake of Fig. 4. There is no wave reflection found at G7.

13. Page 13: Figure 8., y axis would plot only vegetation field, e.g. G3 to G6 and you can define Kv as the vegetation transmission, Kv(x).

Response: We have omitted the original Fig. 8. Instead, we added figures (new Figs. 6 and 7) to discuss the wave height evolution and the deflection angle variation of cylinder for different spring constants.

Technical corrections:

1. Page 2, line 17: Typo. Substitute 'IHFORM' with 'IHFOAM.'

Response: The typo has been corrected.

2. Page 4, line 25: Typo. Substitute 'IHFORM' with 'IHFOAM.'

Response: The typo has been corrected.

3. Page 5, line 17: 'through each cylinder' should be re-written as 'through G3 to G6' and corresponding Fig. 6 needed to increase the font size of G3 to G6. Also, caption needs to be changed as well.

Response: All figures have been corrected.

4. Page 6, line 26: Typo. Substitute 'moving object (GMO)' with 'general moving object (GMO).'

Response: The typo has been corrected.

Please also note the supplement to this comment:
http://www.nat-hazards-earth-syst-sci-discuss.net/nhess-2016-353/nhess-2016-353-AC2-supplement.pdf

**Supplement:**

**Simulations of Swaying Effect of Coastal Vegetation on Tsunami Damping**

Ching-Piao Tsai[1], Ying-Chi Chen[1], Tri Octaviani Sihombing[2] and Chang Lin[1]

[1]Department of Civil Engineering, National Chung Hsing University, Taichung 402, Taiwan
[2]Department of Civil Engineering, Maranatha Christian University, Bandung 40164, Indonesia

*Correspondence to*: Ching-Piao Tsai (cptsai@nchu.edu.tw)

**Abstract.** A coupled wave-vegetation simulation is presented for the swaying effect of the coastal vegetation on tsunami wave height damping. The problem is idealized by solitary wave propagating on a group of emergent cylinders. The numerical model is based on general Reynolds-averaged Navier-Stokes equations with renormalization group turbulent closure model by using volume of fluid technique. The general moving object (GMO) model developed in CFD code Flow-3D is applied to simulate the coupled motion of vegetation with wave dynamically. The damping of wave height and the turbulent kinetic energy as waves passed over both swaying and stationary cylinders are discussed. The simulated results show that the damping of wave height and turbulent kinetic energy by the swaying cylinders were markedly less than by the stationary cylinders. The result implies that the wave decay by the coastal vegetation might be overestimated if the mangrove vegetation was represented as stationary state.

**1 Introduction**

A huge tsunami in South East Asia caused catastrophic damage and claimed more than 200,000 people in December 2004. Cochard et al. (2008) pointed out that this event has stimulated a debate about the role played coastal ecosystems such as mangrove forests and coral reefs in protecting low-lying coastal area. For example, Baird (2006) questioned the effectiveness of the coastal forests or reefs on the reduction of the damage caused by the tsunami. However, Danielsen et al. (2005) reported areas with coastal tree vegetation were markedly less damaged than areas without. Iverson and Prasad (2007) also indicated that developed areas were far more likely to be damaged than forested zones. Several studies (Hiraishi and Harada, 2003; Harada and Kawata, 2004; Teh et al., 2009) have shown that tsunami wave energy, heights and velocities were significantly reduced as the wave propagates through mangrove forests. Nevertheless, Wolanski (2006) has noted that mangroves probably cannot protect the coast against a tsunami wave greater than a threshold level based on some evidence from observations of the Indian Ocean tsunami. Based on the field observations, Shuto (1987) and Yanagisawa et al. (2009) found that single trees or even entire forests could be destroyed through tilting, uprooting, bending or trunk breaking by

tsunami. For tsunami being always present a threat to lives and property along the most coasts of the world, it remains an important for estimating the effectiveness of the coastal vegetation on the tsunami impact.

Many numerical and experimental approaches have been developed in recent years to help understanding the tsunami wave interactions with coastal vegetation. The coastal tree vegetation was idealized by a group of rigid cylinders in most investigations. Huang et al. (2011) performed both experiments and a numerical model by considering solitary wave propagating on emergent rigid cylinders and found that dense cylinders may reduce the wave transmission because of the increased wave energy dissipation into turbulence in cylinders. By using both direct numerical simulation and macroscopic approach, Maza et al. (2015) simulated the interaction of solitary waves with emergent rigid cylinders based on the arrangement of laboratory experiments of Huang et al. (2011). Previous approaches (e.g. Anderson et al., 2011; Huang et al., 2011; Maza et al., 2015; Wu et al., 2016) assumed that the idealized mangrove vegetation is stationary and neglected the plant motion with the wave.

There are several works investigated the hydraulic resistance of coastal vegetation involving the flexible effect of plants. Zhang et al. (2015) pointed out that the prop roots under tidal hydrodynamic loadings in a mangrove environment can be regarded as fairly rigid on account of a large Young's modulus. However, Augustin et al. (2009) indicated that motion of the flexible elements is an important factor on wave attenuation based on flume tests considering both stiff and flexible parameterised tree models under wave action. Husrin (2013) investigated that the trunk of mangrove with its strength properties may behave as a stiff or flexible structure which also governs its relative contribution to the total energy dissipation under tsunami and storm wave action. Coastal pines, one of typical coastal forest vegetation have longer trunk compared to mangroves, Husrin and Oumeraci (2013) indicated that they are more deflected when subject to similar flow velocity compared to mangroves. Husrin et al. (2012) and Strusińska et al. (2013; 2014) examined the tsunami attenuation by coastal vegetation under laboratory conditions for mature mangroves using parameterized trees including flexible tree models. Maza et al. (2013) presented a new numerical model for the interaction of wave and flexible swaying vegetation which couples the flow and the plant motion considering the plant deformation using RANS equation with $k$-$\varepsilon$ turbulent model.

Some mangrove roots and branches at the stage of growing are hanging from the canopy to the flow; it causes the prop roots to oscillate in the water. This study presents the numerical simulation considering vegetation motion coupled with tsunami waves to investigate the wave damping performance. Similar to the experimental work of Kazemi et al. (2015), we model the swaying motion of the vegetation by attaching rigid cylinders to torsional connectors under wave action. This is also a simplified way to represent some movements of mangroves induced by sediment scour, tilting or uprooting states. A direct numerical model based on computational fluid dynamics (CFD) is presented in this paper for simulating the wave damping characteristics including both stationary and swaying vegetation.

**2 Numerical model description**

Among a number of open source CFD codes available, IHFOAM (Higuera et al., 2013; 2014) is specially designed for coastal engineering applications. IHFOAM model was used in Maza et al. (2015) for direct numerical simulation of the solitary wave interacting with the stationary vegetation. Alternatively, the model Flow-3D (Flow Science, Inc., 2012) is applied in this paper to conduct the numerical simulations including vegetation motion under wave action. Flow-3D provides exclusively the FAVOR (fractional area/volumes obstacle representation) technique (Hirt, 1993) and the general moving object (GMO) model that is capable of simulating the rigid body motion dynamically coupled with fluid flow. The FAVOR technique retains rectangular elements with a simple Cartesian grid system and has been shown to be one of the most efficient methods to treat the immersed solid bodies (Xiao, 1999). The free water surface tracking in the model is accomplished by using volume of fluid (VOF) method (Hirt and Nichols, 1981).

Referring to previous literature, the problem is idealized by solitary wave passing on a group of emergent rigid cylinders. Considering the fluid to be incompressible, the continuity and momentum equations for a moving object formulated with area and volume fraction functions are given as

$$\frac{\partial (u_i A_i)}{\partial x_i} = -\frac{\partial V_F}{\partial t}, \tag{1}$$

$$\frac{\partial u_i}{\partial t} + \frac{A_j u_j}{V_F}\frac{\partial u_i}{\partial x_j} = -\frac{1}{\rho}\frac{\partial p}{\partial x_i} + g_i + \frac{1}{\rho V_F}\frac{\partial}{\partial x_j}\left[2A_j[(\mu+\rho\nu_t)S_{ij} - \frac{2}{3}\rho k\delta_{ij})]\right] \tag{2}$$

where $S_{ij} = (\partial u_i / \partial x_j + \partial u_j / \partial x_i)/2$, $V_F$ is the fractional volume open to the flow and $A_j$ is the fraction area for the subscript direction, the subscripts of $i$ and $j$ = 1, 2, 3 represent $x$-, $y$- and $z$- directions, $x_i$ and $x_j$ represent Cartesian coordinates, $u_i$ and $u_j$ are the mean velocity component in subscript direction, $t$ is the time, $p$ is the pressure intensity, $\rho$ is the fluid density, $g$ is the gravitational acceleration, $\mu$ is the absolute viscosity, $\nu_t$ is the eddy viscosity, $k$ is the turbulent kinetic energy, and $\delta_{ij}$ is the Kronecker delta function such that $\delta_{ij}$ = 1 when $i = j$; $\delta_{ij}$ = 0, when $i \neq j$. It is noted that the above governing equations are rendered to standard RANS equations as both $V_F$ and $A$ are set to unity.

The eddy viscosity $\nu_t$ in Eq. (2) takes the form as

$$\nu_t = c_\mu \frac{k^2}{\varepsilon} \tag{3}$$

where $k$ and $\varepsilon$ represent the turbulent kinetic energy and turbulent energy dissipation rate, respectively. $k$ and $\varepsilon$ are related to the effect of space and time distribution of the turbulent motion, which can be solved by a variety of turbulent closure models including one equation model, two equations $k$-$\varepsilon$ model, Renormalization Group method (RNG $k$-$\varepsilon$ model), Large Eddy Simulation (LES), and Shear Stress Transport ($k$-$\omega$ SST) model etc. The RNG $k$-$\varepsilon$ turbulent model was originally

derived by Yokhot and Orszag (1986) based on Renormalization Group methods and improved by Yakhot et al. (1992) with scale expansions for the Reynolds stress and production of dissipation terms. The RNG $k$-$\varepsilon$ model can be a useful turbulence model for practical engineering and scientific calculations (Speziale and Thangam, 1992). Choi et al. (2007) applied RNG $k$-$\varepsilon$ turbulent model to the three-dimensional simulation of tsunami run-up around conical island and demonstrated that it is with computational efficiency and accuracy.  RNG $k$-$\varepsilon$ turbulent model has been proved having reliability for a wider class of flows, thus it is selected to apply in this paper.

Referring to Yakhot et al. (1992), the turbulent transport equations of the RNG $k$-$\varepsilon$ model are expressed as

$$\frac{\partial k}{\partial t} + \frac{u_i A_i}{V_F} \frac{\partial k}{\partial x_i} = P - \varepsilon + \frac{1}{V_F} \frac{\partial}{\partial x_i}\left(\frac{v_t A_i}{\sigma_k} \frac{\partial k}{\partial x_i}\right) \tag{4}$$

$$\frac{\partial \varepsilon}{\partial t} + \frac{u_i A_i}{V_F} \frac{\partial \varepsilon}{\partial x_i} = c_{\varepsilon 1} \frac{\varepsilon}{k} P - c_{\varepsilon 2} \frac{\varepsilon^2}{k} + \frac{1}{V_F} \frac{\partial}{\partial x_i}\left(\frac{v_t A_i}{\sigma_\varepsilon} \frac{\partial \varepsilon}{\partial x_i}\right) \tag{5}$$

where $P$ is the turbulence kinetic energy production given by

$$P = 2v_t S_{ij}\hat{S}_{ij}, \ \hat{S}_{ij} = \frac{1}{V_F}\left(A_j \frac{\partial u_i}{\partial x_j} + A_i \frac{\partial u_j}{\partial u_i}\right) \tag{6}$$

The coefficients are summarized as follows:

$$C_\mu = 0.085, \ C_{\varepsilon 1} = 1.42 - \frac{\alpha(1 - \alpha/\alpha_o)}{1 + \beta\alpha^3}$$
$$C_{\varepsilon 2} = 1.68, \ \sigma_k = 0.7179, \ \sigma_\varepsilon = 0.7179 \tag{7}$$

where $\alpha = Sk/\varepsilon$, $S = (2S_{ij}\hat{S}_{ij})^{1/2}$, $\alpha_o = 4.38$, $\beta = 0.015$.

For coupling the rigid body motion dynamically with fluid flow, the general moving object (GMO) model is adopted here. Comparing with the continuity equation for stationary obstacle problems, $-\partial V_F / \partial t$ in equation (1) is equivalent to an additional volume source term and exists only in mesh cells around the moving object boundary. It can be calculated using

$$-\frac{\partial V_F}{\partial t} = \frac{S_{obj}}{V_{cell}} u_{obj} n_j \tag{8}$$

where $V_{cell}$ is volume of a mesh cell, $S_{obj}$, $n_j$ and $u_{obj}$ are respectively surface area, unit normal vector and velocity of the moving object in the mesh cell. The relative transport equation for the VOF function $F$ is given using

$$\frac{\partial F}{\partial t} + \frac{1}{V_F} \frac{\partial (Fu_i A_i)}{\partial x_i} = -\frac{F}{V_F} \frac{\partial V_F}{\partial t} \tag{9}$$

According to kinematics, general motion of a rigid body can be divided into a translational motion and a rotational motion.

If the cylinder is considered to sway in the *x*-direction accompanied by wave, angular velocity of the swaying cylinder is the only one non-zero component. Then the equations of motion of the cylinder are rendered as

$$T = J\dot{\omega} \tag{10}$$

where $T$, $J$, and $\dot{\omega}$ are total torque, moment of inertia and angular acceleration about the fixed axis. And the velocity of any point G on the swaying cylinder is calculated by $V_G = \omega r_{G/C}$, where $r_{G/C}$ denotes distance from the fixed end C of the cylinder to point G.

In computing the coupling of fluid and rigid body interaction, the velocity and pressure of fluid flow are first solved. The hydrodynamics forces on the rigid body are then obtained and used to calculate the velocity of the rigid body. Then the volume and area fractions are updated according to the new position of the rigid body, and the source term can be calculated using equation (8). The flow field is computed repeatedly until the convergence is achieved. The similar GMO model was well applied for the numerical simulation of the coupled motion of solid body and waves, e.g. Bhinder et al. (2009), Dental et al. (2014), and Zhao et al. (2014).

As for the boundary conditions for solving the governing equations of flow, the normal stress is in equilibrium with the atmospheric pressure while shearing stress is zero on the free surface. All of the solid surfaces were treated using the no-slip boundary condition. The variation of the turbulent energy and the turbulent energy dissipation on the free surface boundary was set as zero in the normal direction. The solution of solitary wave derived from Boussinesq equations was employed as the incident wave.

**3 Validation**

Huang *et al.* (2011) conducted laboratory experiments in a wave flume for the solitary waves interacting with emergent, rigid vegetation. The vegetation was considered as a group of cylinders which were made of Perpex tubes with a uniform outer diameter of 0.01 m. The computations used the same geometric configuration of Huang's laboratory works. The water depth was uniform and equal to $h = 0.15$ m, and the cylinder height was 0.24 m. The arrangement of cylinders shown in Fig. 1 was selected to validate the present numerical simulation. Examples of two vegetation lengths, $L = 1.635$ m and 0.545 m, shown in previous studies are simulated here. The numerical tank was set by 6 m long, 0.55 m wide and 0.3 m height. Note that the verification of the model performance is only implemented by the case of stationary cylinders because the experimental information on the swaying cylinders by solitary waves is unfortunately lacking.

Two different uniform computational meshes around the cylinder field, 0.002 m and 0.001 m respectively, were used to test the numerical accuracy and the sensitivity to grid size. Fig. 2 shows that FAVOR technique resolved successfully the geometry of cylinders using these two computational grids constructed. It indicates that the FAVOR efficiently uses 29 and 17 points to define each cylinder for the mesh of 0 .001 m × 0.001 m and 0.002 m × 0.002 m, respectively.

Fig. 3 shows the comparison of free surface evolution between the present numerical results and experimental measurements for an incident wave height $H_i = 0.05$ m considering the vegetation length $L = 1.635$ m. The results obtained

by the direct simulation using IHFOAM with $k$-$\omega$ SST turbulent model in Maza *et al.* (2015) were also shown in the figure. The comparisons depict that the present numerical results are in a good agreement with the laboratory experiments and previous numerical simulations. The second validation is performed by considering the vegetation length $L = 0.545$ m to compare with the Fig. 14 of Maza *et al.* (2015) using $H_i = 0.05$ m. The simulated result of wave height evolution shown in Fig. 4 depicts in a good agreement with previous numerical results, though the present simulation used different turbulent model. The comparisons shown in Figs. 3 and 4 also demonstrate that there are almost no differences between both computational meshes for the free surface or wave height evolution.

**4 Results and discussion**

The above comparisons demonstrated the present numerical model is capable of simulating accurately the wave evolution by the group cylinders. The following simulations are performed for solitary wave passing through both the stationary and swaying cylinders. The surface elevation evolution, flow field variation, and the turbulent kinetic energy are analysed and compared between both stationary and swaying cylinders. The numerical domain and the arrangement of cylinders used in the following simulations are the same as in previous section. The fine mesh with 0.001 m is used for the following computations.

The swaying cylinders induced by waves are set-up by the general moving object (GMO) model for coupling the cylinder's motion and fluid flow dynamically. Similar to Kazemi et al. (2015), each cylinder end was simplified by attaching a torsion spring connector on the bottom in the model. The use of torsion spring could not completely reproduce the natural bending behaviour of the mangrove tree, but it allows the cylinders to swaying with the passing wave. Peltola et al. (2000) and Husrin (2013) indicated that the deflection angles for a broken trunk may range from $23°$ to $42°$. Too higher value of the specific gravity and lower spring constant used in the present model scale will produce too larger deflection angle of the cylinders and no longer with elastic behaviour. Accordingly, after many numerical tests, the spring constants are set by values of $k_s = 1 - 1.8$ kgw/m with the cylinder's specific gravity of 0.25 to affirm that cylinders can return back to their original position after being hit by waves.

**4.1 Free surface evolution**

The numerical free surface evolutions for the swaying and stationary cylinders respectively are shown in Fig. 5. The spring constant is set by $k_s = 1.0$ kgw/m in this case. It can be seen that the free surface elevation decays rapidly along the cylinder array by stationary cylinders but mildly by swaying cylinders.

Fig. 6 shows the comparison of wave height evolution for swaying cylinders with different spring constants, which can be seen that the results of swaying and stationary cylinders are almost identical for $k_s = 1.8$. Besides, Fig. 7 shows that the larger spring constants the larger maximum deflection angle is. It also shows that the front rows of cylinders get larger deflection. It is noted that the spring constant $k_s = 1.0$ kgw/m is used in the following examples.

Fig. 8 shows the variation of the wave height damping ratio, $H_D = (H - H_i)/H_i$, along the cylinder array for different incident wave heights. It can be seen that the maximum wave height damping ratio of $H_i/H = 0.33$ is approximately 26% for swaying cylinders but it could reach to 61% for stationary cylinders. The result of free surface evolution depicts that the stationary cylinders are working better than the swaying cylinders for the wave height damping. That is, the wave height decay can be overestimated if the coastal vegetation was considered as stationary state.

**4.2 Flow field evolution**

Fig. 9 shows the snapshots of velocity distribution at the centre line of the tank for swaying and stationary cylinders as the solitary wave crest is passing through gauges G3 to G6 for an incident wave height $H_i = 0.05$ m. It can be observed that the water velocity reduces rapidly along the array of stationary cylinders, but it reduces gently by swaying cylinders. The swaying cylinders have angular motion and even become immerse in the water under the wave crest, thus it leads to the flow velocity be larger than that of stationary cylinders. Fig. 10 shows the comparison of the horizontal velocity profile as wave crest is passing through gauges G3 to G6, i.e. $x/L = 0$, 0.33, 0.66 and 1.0, for swaying and stationary cylinders. It can be seen that the profiles vary oscillatory for swaying cylinders due to the motion effect of cylinder.

**4.3 Turbulent kinetic energy evolution**

The turbulent kinetic energy will be generated and dissipated during the wave interacting with the group of cylinders. The turbulent kinetic energy ($k$) and the turbulent kinetic energy dissipation rate ($\varepsilon$) are obtained from the RNG $k$-$\varepsilon$ turbulent closure model while the general RANS equations is solving. We focus on when and where the maximum turbulent kinetic energy occurs for an incident wave height $H_i = 0.05$ m.

Figs. 11 and 12 display the snapshots of the spatial distribution of the turbulent kinetic energy (TKE) and the turbulent kinetic energy dissipation rate (DTKE) for swaying and stationary cylinders, respectively, when the wave crest is passing through gauges G3 to G6. It shows that the turbulent kinetic energy start generating and dissipating as the wave crest is hitting on the front row of cylinders. It can be seen that the characteristics of spatial distribution of TKE and DTKE for swaying and stationary cylinders are very similar. Figs. 13 and 14 display the time variations of TKE at each section ($x/L = 0-1$), which shows that the maximum TKE occurs at $x/L = 0.33$ of both cylinders. The result can be stated that the maximum TKE is not occurring when the wave crest is reaching the cylinders ($x/L = 0$). This result is similar to Maza *et al.* (2015), which obtained that the maximum turbulent intensity is not developing when the wave crest is reaching the cylinders.

Fig. 13 also shows that there is a time lag between the occurrence of maximum TKE and the maximum wave elevation of stationary cylinders, but there is no lag for swaying cylinders. That is, the maximum TKE is produced after wave crest passed each section of stationary cylinders, but it is produced as wave crest is passing each section for swaying cylinders. However, it can also be found from Fig. 14 that multiple peaks of the TKE evolution exist in the case of swaying cylinders by the return back process to its original position.

Fig. 15 shows the comparisons of vertical profile of TKE between swaying and stationary cylinders as the wave crest is passing through gauges G3 to G6. It can be seen that the strongest shear layer of both cylinders is generated near the free surface where the largest turbulence occurs, and the TKE decreases along the cylinder array. We can find that the vertical profile of TKE exists multiple shear layers due to the swaying effect of the cylinders. Fig. 15 shows the comparisons of total TKE evolution along the array of both cylinders, in which the total TKE is calculated by the integral of time evolution shown in Figs. 13 and 14. It can be found that the larger TKE occurs between $x/L = 0.165$ and $x/L = 0.495$ for both cylinders and the TKE of swaying cylinders is obviously less than that of stationary cylinders.

**5. Conclusions**

A numerical simulation based on the general RANS equations and RNG $k$-$\varepsilon$ turbulent model was implemented to investigate the swaying effect of coastal vegetation on the damping of tsunami wave. The vegetation was idealized by a group of emergent, rigid cylinders. The FAVOR technique and general moving object (GMO) model provided in Flow-3D code were employed in this paper for simulating the coupling of fluid and rigid body interaction. The evolutions of wave height, flow field and turbulent kinetic energy for both stationary and swaying cylinder are investigated. Due to the swaying effect of the cylinders under the wave action, the numerical results showed that the damping of wave height and turbulent kinetic energy were markedly less than those of stationary cylinders. That is, tsunami damping can be overestimated if the coastal vegetation is represented as a stationary state.

**Acknowledgements**

The authors would like to express their deep appreciation to Professor Maza and the anonymous reviewer for their invaluable comments and suggestions, which have led us to improve the quality of the paper.

**References**

Anderson, M.E., Smith, J.M., and McKay, S.K.: Wave dissipation by vegetation, Coastal and Hydraulics Engineering Technical Note ERDC/CHL CHETN-I-82, U.S. Army Engineer Research and Development Center, Vicksburg, MS., 2011.

Augustin, L.N., Irish, J.L., and Lynett, P.: Laboratory and numerical studies of wave damping by emergent and near-emergent wetland vegetation, Coast. Eng., 56, 332-340, 2009.

Baird, A.: False hopes and natural disasters, New York Times, 26 December 2006.

Bhinder, M.A., Mingham, C.G., Causon, D.M., Rahmati, M.T., Aggidis, G.A., and Chaplin, R.V.: A joint numerical and experimental study of a surging point absorbing wave energy of converter (WRASPA), Proc. ASME 28th Inter. Conf. Ocean, Offshore and Arctic Eng., OMAE2009-79392, 2009.

Choi, B.H., Kim, D.C., Pelinovsky, E., and Woo, S.B.: Three-dimensional simulation of tsunami run-up around conical island. Coast. Eng., 54, 618–629, 2007.

Cochard, R., Ranamukhaarachchi, S.L., Shivakoti, G.P., Shipin, O.V., Edwards, P.J., and Seeland, K. T. The 2004 tsunami in Aceh and Southern Thailand: A review on coastal ecosystems, wave hazards and vulnerability, Perspectives in plant ecology, evolution and systematics, 10(1), 3-40, 2008.

Danielsen, F., Sorensen, M.K., Olwig, M.F., Selvam, V., Parish, F., Burgess,N.D., Hiraishi, T., Karunagaran,V .M., Rasmussen, M.S., Hansen, L.B., Quarto, A., and Suryadiputra, N.: The Asian tsunami: a protective role for coastal vegetation, Science, 310, 643, 2005.

Dentale, F., Donnarumma, G., and Pugliese Carratelli, E.: Simulation of flow within armour blocks in a breakwater. J. Coast. Res., 30(3), 528-536, 2014.

Flow Science, Inc.: Flow-3D User's Manuals. Santa Fe, NM, 2012.

Harada, K. and Kawata, Y.: Study on the effect of coastal forest to tsunami reduction, Annuals of Disas. Prev. Res. Inst., Kyoto Univ., No. 47 C, 2004.

Higuera, P., Lara, J.L., and Losada, I.J.: Realistic wave generation and active wave absorption for Navier–Stokes models: application to OpenFOAM, Coast. Eng., 71, 102–118, 2013.

Higuera, P., Lara, J.L., and Losada, I.J.: Three-dimensional interaction of waves and porous coastal structures using OpenFOAM. Part I: formulation and validation. Coast. Eng., 83, 243–258, 2014.

Hiraishi, T. and Harada, K.: Greenbelt tsunami prevention in south-Pacific region, Report of the Port and Airport Research Institute, 42 (2), 3–25, 2003.

Hirt, C.W. and Nichols, B.D.: Volume of fluid (VOF) method for dynamics of free boundaries, J. Comp. Phys., 39, 201-225, 1981.

Hirt, C. W.: Volume-fraction techniques: powerful tools for wind engineering, J. Wind Eng. Indust. Aerodynamics, 46-47, 327–338, 1993.

Huang, Z., Yao, Y., Sim, S.Y., and Yao, Y.: Interaction of solitary waves with emergent stationary vegetation, Ocean Eng., 38, 1080–1088, 2011.

Husrin, S., Strusi´nska, A., and Oumeraci, H.: Experimental study on tsunami attenuation by mangrove forest, Earth, Planets Space, 64, 973–989, 2012.

Husrin, S.: Attenuation of solitary waves and wave trains by coastal forests, Doctoral Thesis, TU Braunschweig, Germany, 2013.

Iverson, L.R. and Prasad, A.M.: Using landscape analysis to assess and model tsunami damage in Aceh province, Sumatra, Landscape Ecol., 22, 323-331, 2007.

Kazemi, A., Parry, S., van de Riet, K, and Curet, O.: The effect of porosity and flexibility on the hydrodynamics behind a mangrove-like root model, APS Division of Fluid Dynamics (Fall) 2015, abstract #A25.007, 2015.

Maza, Maria, Lara, J.L., and Losada, I.J.: Tsunami wave interaction with mangrove forests: a 3-D numerical approach, Coast. Eng., 98, 33-54, 2015.

Peltola, H., Kellomäki, S., Hassinen, A., and Granander, M.,: 'Mechanical stability of Scots pine, Norway spruce and birch: an analysis of tree-pulling experiments in Finland', Forest Ecology and Management, 135, 143-153, 2000.

5  Shuto, N.: The effectiveness and limit of tsunami control forests, Coastal Engineering in Japan, 30, 143–153, 1987.

Strusińska, A.; Husrin, S., and Oumeraci, H.: Tsunami damping by mangrove forest: a laboratory study using parameterized trees, Nat. Hazards Earth Syst. Sci., 13, 483–503, 2013.

Strusińska, A.; Husrin, S., and Oumeraci, H.: Attenuation of solitary wave by parameterized flexible mangrove model, Proc. 34th Intl. Conf. Coast. Eng., 2014.

10  Teh, S.Y., Koh, H.L., Liu, P.L.F., Ismail, A.I.M., and Lee, H.L.: Analytical and numerical simulation of tsunami mitigation by mangroves in Penang, Malaysia, J. Asian Earth Sci., 36, 38–46, 2009.

Wolanski, E.: Synthesis of the protective functions of coastal forests and trees against natural hazards, Proceedings of the regional technical workshop on Coastal protection in the aftermath of the Indian Ocean tsunami: What role for forests and trees?, Food and Agriculture Organization of the United Nations, 2006.

15  Wu, W.C., Ma G., and Cox, D.T.: Modeling wave attenuation induced by the vertical density variations of vegetation, Coast. Eng., 112, 17-27, 2016.

Xiao, F.: A computational model for suspended large rigid bodies in 3D unsteady viscous flows. J. Comput. Phys., 155, 348–379, 1999.

Yakhot, V. and Orszag, S. A.: Renormalization group analysis of turbulence, I. Basic theory, J. Sci. Comput. 1(1), 3-51, 1986.

Yakhot, V., Orszag, S.A., Thangam, S., Gatski, T.B., and Speziale, C.G.: Development of turbrulence models for shear flows by a double expansion technique, Phys. Fluids A, 4(7), 1510-1520., 1992.

Zhang, X, Chua, V.P., and Cheong, H.F.: Hydrodynamics in mangrove prop roots and their physical properties, J. Hydro-environ. Res., 9. 281-294, 2015.

25  Zhao, X., Ye, Z., Fu, Y., and Cao, F.: A CIP-based numerical simulation of freak wave impact on a floating body, Ocean Eng., 87, 50–63, 2014.

30

0.06 m

0.06
m

**Figure 1.** Cylinder cell arrangement (left), field length (right) and locations of wave probes for the computations.

[Figure]

**Figure 2.** FAVORized geometry of cylinders and constructed computational rectangular grid with 0.001 m (left) and 0.002 m (right).

[Figure]

**Figure 3.** Comparison of free surface evolution between numerical and experimental results for $H_i = 0.05$ m.

[Figure]

**Figure 4.** Comparison between the numerical results for the wave height for $H_i$ = 0.05 m using the field length 0.545 m.

[Figure]

[Figure]

**Figure 5**. Wave surface evolutions of the swaying and stationary cylinders for an incident wave height $H_i$ = 0.05 m, $H_i/h$ = 0.33.

[Figure]

**Figure 6.** Comparison of wave height evolutions for swaying cylinders with different spring constants.

[Figure]

**Figure 7.** Comparison of the maximum deflection angles of swaying cylinders with different spring constants.

[Figure]

**Figure 8.** Comparison of wave height damping ratio between swaying and stationary cylinders for different incident wave heights.

[Figure]

**Figure 9.** The snapshots of the velocity distribution for the stationary cylinders (left) and swaying cylinders (right) for $H_i$ = 0.05 m.

[Figure]

**Figure 10.** Comparison of the horizontal velocity profile for the swaying and stationary cylinders as wave crest is passing each section for $H_i = 0.05$ m.

[Figure]

**Figure 11.** Snapshots of TKE for stationary cylinders (left) and swaying cylinders (right) for $H_i = 0.05$ m.

[Figure]

**Figure 12.** Snapshots of DTKE for stationary cylinders (left) and swaying cylinders (right) for $H_i$ = 0.05 m.

[Figure]

**Figure 13.** The time evolution of TKE at each section for stationary cylinders for $H_i = 0.05$ m.

[Figure]

**Figure 14.** The time evolution of TKE at each section for swaying cylinders for $H_i = 0.05$ m.

[Figure]

**Figure 15.** Comparison of vertical profiles of TKE as wave crest is passing through each section for $H_i = 0.05$ m.

[Figure]

**Figure 16.** Comparison of total TKE evolution along cylinder array for $H_i = 0.05$ m.